# A Kernel Loss for Solving the Bellman Equation

**Yihao Feng**
UT Austin
yihao@cs.utexas.edu

**Lihong Li**
Google Research
lihong@google.com

**Qiang Liu**
UT Austin
lqiang@cs.utexas.edu

## Abstract

Value function learning plays a central role in many state-of-the-art reinforcement-learning algorithms. Many popular algorithms like Q-learning do not optimize any objective function, but are fixed-point iterations of some variants of Bellman operator that are not necessarily a contraction. As a result, they may easily lose convergence guarantees, as can be observed in practice. In this paper, we propose a novel loss function, which can be optimized using standard gradient-based methods with guaranteed convergence. The key advantage is that its gradient can be easily approximated using sampled transitions, avoiding the need for double samples required by prior algorithms like residual gradient. Our approach may be combined with general function classes such as neural networks, using either on- or off-policy data, and is shown to work reliably and effectively in several benchmarks, including classic problems where standard algorithms are known to diverge.

## 1 Introduction

The goal of a reinforcement learning (RL) agent is to optimize its policy to maximize the long-term return through repeated interaction with an external environment. The interaction is often modeled as a Markov decision process, whose value functions are the unique fixed points of their corresponding *Bellman operators*. Many state-of-the-art algorithms, including TD($\lambda$), Q-learning and actor-critic, have value function learning as a key component (Sutton & Barto, 2018; Szepesvári, 2010).

A fundamental property of the Bellman operator is that it is a contraction in the value function space in the $\ell_\infty$-norm (Puterman, 1994). Therefore, starting from any bounded initial function, with repeated applications of the operator, the value function converges to the true value function. A number of algorithms are directly inspired by this property, such as temporal difference (Sutton, 1988) and its many variants (Bertsekas & Tsitsiklis, 1996; Sutton & Barto, 2018; Szepesvári, 2010). Unfortunately, when function approximation such as neural networks is used to represent the value function in large-scale problems, the critical property of contraction is generally lost (e.g., Boyan & Moore, 1995; Baird, 1995; Tsitsiklis & Van Roy, 1997), except in rather restricted cases (e.g., Gordon, 1995; Tsitsiklis & Van Roy, 1997). Not only is this instability one of the core theoretical challenges in RL, but it also has broad practical significance, given the growing popularity of algorithms like DQN (Mnih et al., 2015), A3C (Mnih et al., 2016) and their many variants (e.g., Gu et al., 2016; Schulman et al., 2016; Wang et al., 2016; Wu et al., 2017), whose stability largely depends on the contraction property. The instability becomes even harder to avoid, when training data (transitions) are sampled from an off-policy distribution, a situation known as the *deadly triad* (Sutton & Barto, 2018, Sec. 11.3).

The brittleness of Bellman operator's contraction property has inspired a number of works that aim to reformulate value function learning as an optimization problem, where standard algorithms like stochastic gradient descent can be used to minimize the objective, without the risk of divergence

(under mild and typical assumptions). One of the earliest attempts is residual gradient, or RG (Baird, 1995), which relies on minimizing squared temporal differences. The algorithm is convergent, but its objective is not necessarily a good proxy due to a well-known "double sample" problem. As a result, it may converge to an inferior solution; see Sections 2 and 6 for further details and numerical examples. This drawback is inherited by similar algorithms like PCL (Nachum et al., 2017, 2018).

Another line of work seeks alternative objective functions, the minimization of which leads to desired value functions (e.g., Sutton et al., 2009; Maei, 2011; Liu et al., 2015; Dai et al., 2017). Most existing works are either for linear approximation, or for evaluation of a fixed policy. An exception is the SBEED algorithm (Dai et al., 2018b), which transforms the Bellman equation to an equivalent saddle-point problem, and can use nonlinear function approximations. While SBEED is provably convergent under fairly standard conditions, it relies on solving a minimax problem, whose optimization can be rather challenging in practice, especially with nonconvex approximation classes like neural networks.

In this paper, we propose a novel loss function for value function learning. It avoids the double-sample problem (unlike RG), and can be easily estimated and optimized using sampled transitions (in both on- and off-policy scenarios). This is made possible by leveraging an important property of integrally strictly positive definite kernels (Stewart, 1976; Sriperumbudur et al., 2010). This new objective function allows us to derive simple yet effective algorithms to approximate the value function, without risking instability or divergence (unlike TD algorithms), or solving a more sophisticated saddle-point problem (unlike SBEED). Our approach also allows great flexibility in choosing the value function approximation classes, including nonlinear ones like neural networks. Experiments in several benchmarks demonstrate the effectiveness of our method, for both policy evaluation and optimization problems. We will focus on the batch setting (or the growing-batch setting with a growing replay buffer), and leave the online setting for future work.

## 2 Background

This section starts with necessary notation and background information, then reviews two representative algorithms that work with general, nonlinear (differentiable) function classes.

**Notation.** A Markov decision process (MDP) is denoted by $M = \langle \mathcal{S}, \mathcal{A}, P, R, \gamma \rangle$, where $\mathcal{S}$ is a (possibly infinite) state space, $\mathcal{A}$ an action space, $P(s' \mid s, a)$ the transition probability, $R(s, a)$ the average immediate reward, and $\gamma \in (0, 1)$ a discount factor. The value function of a policy $\pi : \mathcal{S} \mapsto \mathbb{R}_+^{\mathcal{A}}$, denoted $V^\pi(s) := \mathbb{E}\left[\sum_{t=0}^\infty \gamma^t R(s_t, a_t) \mid s_0 = s, a_t \sim \pi(\cdot, s_t)\right]$, measures the expected long-term return of a state. It is well-known that $V = V^\pi$ is the unique solution to the *Bellman equation* (Puterman, 1994), $V = \mathcal{B}_\pi V$, where $\mathcal{B}_\pi : \mathbb{R}^{\mathcal{S}} \to \mathbb{R}^{\mathcal{S}}$ is the *Bellman operator*, defined by

$$\mathcal{B}_\pi V(s) := \mathbb{E}_{a \sim \pi(\cdot|s), s' \sim P(\cdot|s,a)}[R(s, a) + \gamma V(s') \mid s].$$

While we develop and analyze our approach mostly for $\mathcal{B}_\pi$ given a fixed $\pi$ (policy evaluation), we will also extend the approach to the controlled case of policy optimization, where the corresponding Bellman operator becomes

$$\mathcal{B}V(s) := \max_a \mathbb{E}_{s' \sim P(\cdot|s,a)}[R(s, a) + \gamma V(s') \mid s, a].$$

The unique fixed point of $\mathcal{B}$ is known as the optimal value function, denoted $V^*$; that is, $\mathcal{B}V^* = V^*$.

Our work is built on top of an alternative to the fixed-point view above: given some fixed distribution $\mu$ whose support is $\mathcal{S}$, $V^\pi$ is the unique minimizer of the *squared Bellman error*:

$$L_2(V) := \|\mathcal{B}_\pi V - V\|_\mu^2 = \mathbb{E}_{s \sim \mu}\left[(\mathcal{B}_\pi V(s) - V(s))^2\right].$$

Denote by $\mathcal{R}_\pi V := \mathcal{B}_\pi V - V$ the Bellman error operator. With a set $\mathcal{D} = \{(s_i, a_i, r_i, s_i')\}_{1 \le i \le n}$ of transitions where $a_i \sim \pi(\cdot|s_i)$, the Bellman operator in state $s_i$ can be approximated by *bootstrapping*: $\hat{\mathcal{B}}_\pi V(s_i) := r_i + \gamma V(s_i')$. Similarly, $\hat{\mathcal{R}}_\pi V(s_i) := r_i + \gamma V(s_i') - V(s_i)$. Clearly, one has $\mathbb{E}[\hat{\mathcal{B}}_\pi V(s_i)|s_i] = \mathcal{B}_\pi V(s_i)$ and $\mathbb{E}[\hat{\mathcal{R}}_\pi V(s_i)|s_i] = \mathcal{R}_\pi V(s_i)$. In the literature, $\hat{\mathcal{R}}_\pi V(s_i)$ is also known as the temporal difference or TD error, whose expectation is the Bellman error.

Finally, in this work, we use the same notation for a distribution and its probability density function.

**Basic Algorithms.** We are interested in estimating $V^\pi$, from a parametric family $\{V_\theta : \theta \in \Theta\}$, from data $\mathcal{D}$. The *residual gradient* algorithm (Baird, 1995) minimizes the *squared TD error*:

$$\hat{L}_{\mathrm{RG}}(V_\theta) := \frac{1}{n} \sum_{i=1}^{n} \left( \hat{\mathcal{B}}_\pi V_\theta(s_i) - V_\theta(s_i) \right)^2, \tag{1}$$

with gradient descent update $\theta_{t+1} = \theta_t - \epsilon \nabla_\theta \hat{L}_{\mathrm{RG}}(V_{\theta_t})$, where

$$\nabla_\theta \hat{L}_{\mathrm{RG}}(V_\theta) = \frac{2}{n} \sum_{i=1}^{n} \left( \left( \hat{\mathcal{B}}_\pi V_\theta(s_i) - V_\theta(s_i) \right) \cdot \nabla_\theta \left( \hat{\mathcal{B}}_\pi V_\theta(s_i) - V_\theta(s_i) \right) \right).$$

However, the objective in (1) is a biased and inconsistent estimate of the squared Bellman error. This is because $\mathbb{E}_{s\sim\mu}[\hat{L}_{\mathrm{RG}}(V)] = L_2(V) + \mathbb{E}_{s\sim\mu}[\mathrm{var}(\hat{\mathcal{B}}_\pi V(s)|s)] \neq L_2(V)$, where there is an extra term that involves the conditional variance of the empirical Bellman operator, which does not vanish unless the state transitions are deterministic. As a result, RG can converge to incorrect value functions (see also Section 6). With random transitions, correcting the bias requires double samples (i.e., at least two independent samples of $(r, s')$ for the same $(s, a)$ pair) to estimate the conditional variance.

More popular algorithms in the literature are instead based on fixed-point iterations, using $\hat{\mathcal{B}}_\pi$ to construct a target value to update $V_\theta(s_i)$. An example is *fitted value iteration*, or FVI (Bertsekas & Tsitsiklis, 1996; Munos & Szepesvári, 2008), which includes as special cases the empirically successful DQN and variants, and also serves as a key component in many state-of-the-art actor-critic algorithms. In its basic form, FVI starts from an initial $\theta_0$, and iteratively updates the parameter by

$$\theta_{t+1} = \operatorname*{arg\,min}_{\theta \in \Theta} \left\{ \hat{L}_{\mathrm{FVI}}^{(t+1)}(V_\theta) := \frac{1}{n} \sum_{i=1}^{n} \left( V_\theta(s_i) - \hat{\mathcal{B}}_\pi V_{\theta_t}(s_i) \right)^2 \right\}. \tag{2}$$

Different from RG, when gradient-based methods are applied to solve (2), the current parameter $\theta_t$ is treated as a constant: $\nabla_\theta \hat{L}_{\mathrm{FVI}}^{(t+1)}(V_\theta) = \frac{2}{n} \sum_{i=1}^{n} \left( V_\theta(s_i) - \hat{\mathcal{B}}_\pi V_{\theta_t}(s_i) \right) \nabla_\theta V_\theta(s_i)$. TD(0) (Sutton, 1988) may be viewed as a stochastic version of FVI, where a single sample (i.e., $n = 1$) is drawn randomly (either from a stream of transitions or from a replay buffer) to estimate the gradient of (2).

Being fixed-point iteration methods, FVI-style algorithms *do not* optimize any objective function, and their convergence is guaranteed only in rather restricted cases (e.g., Gordon, 1995; Tsitsiklis & Van Roy, 1997; Antos et al., 2008). Such divergent behavior is well-known and empirically observed (Baird, 1995; Boyan & Moore, 1995); see Section 6 for more numerical examples. It creates substantial difficulty in parameter tuning and model selection in practice.

## 3 Kernel Loss for Policy Evaluation

Much of the algorithmic challenge described earlier lies in the difficulty in estimating squared Bellman error from data. In this section, we address this difficulty by proposing a new loss function that is more amenable to statistical estimation from empirical data. Proofs are deferred to the appendix.

Our framework relies on an *integrally strictly positive definite (ISPD) kernel* $K : \mathcal{S} \times \mathcal{S} \to \mathbb{R}$, which is a symmetric bi-variate function that satisfies $\|f\|_K^2 := \int_{\mathcal{S}^2} K(s, \bar{s}) f(s) f(\bar{s}) \, ds \, d\bar{s} > 0$, for any non-zero $L_2$-integrable function $f$. For simplicity, we consider two functions $f$ and $g$ equal if $(f - g)$ has a zero $L_2$ norm. We call $\|f\|_K$ the $K$-norm of $f$. Many commonly used kernels, such as Gaussian RBF kernel $K(s, \bar{s}) = \exp(-\|s - \bar{s}\|_2^2 / h)$ is ISPD. More discussion on ISPD kernels can be found in Stewart (1976) and Sriperumbudur et al. (2010).

### 3.1 The New Loss Function

Recall that $\mathcal{R}_\pi V = \mathcal{B}_\pi V - V$ is the Bellman error operator. Our new loss function is defined by

$$L_K(V) = \|\mathcal{R}_\pi V\|_{K,\mu}^2 := \mathbb{E}_{s,\bar{s}\sim\mu} \left[ K(s, \bar{s}) \cdot \mathcal{R}_\pi V(s) \cdot \mathcal{R}_\pi V(\bar{s}) \right], \tag{3}$$

where $\mu$ is any positive density function on states $s$, and $s, \bar{s} \sim \mu$ means $s$ and $\bar{s}$ are drawn i.i.d. from $\mu$. Here, $\|\cdot\|_{K,\mu}$ is regarded as the $K$-norm under measure $\mu$. It is easy to show that $\|f\|_{K,\mu} = \|f\mu\|_K$. Note that $\mu$ can be either the visitation distribution under policy $\pi$ (the *on-policy* case), or some other distribution (the *off-policy* case). Our approach handles both cases in a unified way. The following theorem shows that the loss $L_K$ is consistent:

**Theorem 3.1.** *Let $K$ be an ISPD kernel and assume $\mu(s) > 0, \forall s \in \mathcal{S}$. Then, $L_K(V) \geq 0$ for any $V$; and $L_K(V) = 0$ if and only if $V = V^\pi$. In other words, $V^\pi = \arg\min_V L_K(V)$.*

The next result relates the kernel loss to a "dual" kernel norm of the value function error, $V - V^\pi$.

**Theorem 3.2.** *Under the same assumptions as Theorem 3.1, we have $L_K(V) = \|V - V^\pi\|^2_{K^*,\mu}$, where $\|\cdot\|_{K^*,\mu}$ is the $K^*$-norm under measure $\mu$ with a "dual" kernel $K^*(s, \bar{s})$, defined by*

$$K^*(s', \bar{s}') := \mathbb{E}_{s,\bar{s} \sim d^*_{\pi,\mu}} \left[ K(s', \bar{s}') + \gamma^2 K(s, \bar{s}) - \gamma \big( K(s', \bar{s}) + K(s, \bar{s}') \big) \mid s', \bar{s}' \right],$$

*and the expectation notation is shorthand for $\mathbb{E}_{s \sim d^*_{\pi,\mu}}[f(s)|s'] = \int f(s) d^*_{\pi,\mu}(s|s')ds$, with*

$$d^*_{\pi,\mu}(s|s') := \sum_a \pi(a|s)P(s'|s,a)\mu(s)/\mu(s').$$

The norm involves a quantity, $d^*_{\pi,\mu}(s|s')$, which may be heuristically viewed as a "backward" conditional probability of state $s$ conditioning on observing the next state $s'$ (but note that $d^*_{\pi,\mu}(s|s')$ is not normalized to sum to one unless $\mu = d_\pi$).

**Empirical Estimation** The key advantage of the new loss $L_K$ is that it can be easily estimated and optimized from observed transitions, without requiring double samples. Given a set of empirical data $\mathcal{D} = \{(s_i, a_i, r_i, s'_i)\}_{1 \leq i \leq n}$, a way to estimate $L_K$ is to use the so-called *V-statistics*,

$$\hat{L}_K(V_\theta) := \frac{1}{n^2} \sum_{1 \leq i,j \leq n} K(s_i, s_j) \cdot \hat{\mathcal{R}}_\pi V_\theta(s_i) \cdot \hat{\mathcal{R}}_\pi V_\theta(s_j). \tag{4}$$

Similarly, the gradient $\nabla_\theta L_K(V_\theta) = 2\mathbb{E}_\mu[K(s, \bar{s})\mathcal{R}_\pi V_\theta(s)\nabla_\theta(\mathcal{R}_\pi V_\theta(\bar{s}))]$ can be estimated by

$$\nabla_\theta \hat{L}_K(V_\theta) := \frac{2}{n^2} \sum_{1 \leq i,j \leq n} K(s_i, s_j) \cdot \hat{\mathcal{R}}_\pi V_\theta(s_i) \cdot \nabla_\theta \hat{\mathcal{R}}_\pi V_\theta(s_j).$$

Note that while calculating the exact gradient requires $O(n^2)$ computation, in practice we may use stochastic gradient descent on mini-batches of data instead. The precise formulas for unbiased estimates of the gradient of the kernel loss using a subset of samples are given in Appendix B.1.

**Remark** (unbiasedness) An alternative approach is to use the *U-statistics*, which removes the diagonal ($i = j$) terms in the pairwise average in (4). In the case of i.i.d. samples, it is known that U-statistics forms an unbiased estimate of the true gradient, but may have higher variance than the V-statistics. In our experiments, we observe that V-statistics works better than U-statistics.

**Remark** (consistency) Following standard statistical approximation theory (e.g., Serfling, 2009), both U/V-statistics provide *consistent* estimation of the expected quadratic quantity given the sample is weakly dependent and satisfies certain mixing condition (e.g., Denker & Keller, 1983; Beutner & Zähle, 2012); this often amounts to saying that $\{s_i\}$ forms a Markov chain that converges to its stationary distribution $\mu$ sufficiently fast. This is in contrast to the gradient computed by residual gradient, which is known to be inconsistent in general.

**Remark** Another advantage of our kernel loss is that we have $L_K(V) = 0$ iff $V = V^\pi$. Therefore, the magnitude of the empirical loss $\hat{L}_K(V)$ reflects the closeness of $V$ to the true value function $V^\pi$. In fact, by using methods from kernel-based hypothesis testing (e.g., Gretton et al., 2012; Liu et al., 2016; Chwialkowski et al., 2016), one can design statistically calibrated methods to test if $V = V^\pi$ has been achieved, which may be useful for designing efficient exploration strategies. In this work, we focus on estimating $V^\pi$ and leave it as future work to test value function proximity.

## 3.2 Interpretations of the Kernel Loss

We now provide some insights into the new loss function, based on two interpretations.

**Eigenfunction Interpretation**   Mercer's theorem implies the following decomposition

$$K(s, \bar{s}) = \sum_{i=1}^{\infty} \lambda_i e_i(s) e_i(\bar{s}) \,, \tag{5}$$

of any continuous positive definite kernel on a compact domain, where $\{e_i\}_{i=1}^{\infty}$ is a countable set of orthonormal eigenfunctions w.r.t. $\mu$ (i.e., $\mathbb{E}_{s \sim \mu}[e_i(s) e_j(s)] = \mathbf{1}\{i = j\}$), and $\{\lambda_i\}_{i=1}^{\infty}$ are their eigenvalues. For ISPD kernels, all the eigenvalues must be positive: $\forall i, \, \lambda_i > 0$.

The following shows that $L_K$ is a squared projected Bellman error in the space spanned by $\{e_i\}_{i=1}^{\infty}$.

**Proposition 3.3.** *If* (5) *holds, then*

$$L_K(V) = \sum_{i=1}^{\infty} \lambda_i \left( \mathbb{E}_{s \sim \mu} \left[ \mathcal{R}_\pi V(s) \times e_i(s) \right] \right)^2 \,.$$

*Moreover, if* $\{e_i\}$ *is a complete orthonormal basis of* $L_2$-*space under measure* $\mu$, *then the* $L_2$ *loss is*

$$L_2(V) = \sum_{i=1}^{\infty} \left( \mathbb{E}_{s \sim \mu} \left[ \mathcal{R}_\pi V(s) \times e_i(s) \right] \right)^2 \,.$$

*Therefore,* $L_K(V) \leq \lambda_{\max} L_2(V)$, *where* $\lambda_{\max} := \max_i \{ \lambda_i \}$.

This result shows that the eigenvalue $\lambda_i$ controls the contribution of the projected Bellman error to the eigenfunction $e_i$ in $L_K$. It may be tempting to have $\lambda_i \equiv 1$, in which $L_K(V) = L_2(V)$, but the Mercer expansion in (5) can diverge to infinity, resulting in an ill-defined kernel $K(s, \bar{s})$. To avoid this, the eigenvalues must decay to zero fast enough such that $\sum_{i=1}^{\infty} \lambda_i < \infty$. Therefore, the kernel loss $L_K(V)$ can be viewed as prioritizing the projections to the eigenfunctions with larger eigenvalues. In typical kernels such as Gaussian RBF kernels, these dominant eigenfunctions are Fourier bases with low frequencies (and hence high smoothness), which may intuitively be more relevant than the higher frequency bases for practical purposes.

**RKHS Interpretation**   The squared Bellman error has the following variational form:

$$L_2(V) = \max_f \left\{ \left( \mathbb{E}_{s \sim \mu} \left[ \mathcal{R}_\pi V(s) \times f(s) \right] \right)^2 : \quad \mathbb{E}_{s \sim \mu}[(f(s))^2] \leq 1 \right\} \,, \tag{6}$$

which involves finding a function $f$ in the unit $L_2$-ball, whose inner product with $\mathcal{R}_\pi V(s)$ is maximal. Our kernel loss has a similar interpretation, with a different unit ball.

Any positive kernel $K(s, \bar{s})$ is associated with a Reproducing Kernel Hilbert Space (RKHS) $\mathcal{H}_K$, which is the Hilbert space consisting of (the closure of) the linear span of $K(\cdot, s)$, for $s \in \mathcal{S}$, and satisfies the reproducing property, $f(x) = \langle f, K(\cdot, x) \rangle_{\mathcal{H}_K}$, for any $f \in \mathcal{H}_K$. RKHS has been widely used as a powerful tool in various machine learning and statistical problems; see Berlinet & Thomas-Agnan (2011); Muandet et al. (2017) for overviews.

**Proposition 3.4.** *Let* $\mathcal{H}_K$ *be the RKHS of kernel* $K(s, \bar{s})$, *we have*

$$L_K(V) = \max_{f \in \mathcal{H}_K} \left\{ \left( \mathbb{E}_{s \sim \mu} \left[ \mathcal{R}_\pi V(s) \times f(s) \right] \right)^2 : \quad \| f \|_{\mathcal{H}_K} \leq 1 \right\}. \tag{7}$$

Since RKHS is a subset of the $L_2$ space that includes smooth functions, we can again see that $L_K(V)$ emphasizes more the projections to smooth basis functions, matching the intuitive from Theorem 3.3. It also draws a connection to the recent primal-dual reformulations of the Bellman equation (Dai et al., 2017, 2018b), which formulate $V^\pi$ as a saddle-point of the following minimax problem:

$$\min_V \max_f \; \mathbb{E}_{s \sim \mu} \left[ 2 \mathcal{R}_\pi V(s) \times f(s) - f(s)^2 \right] . \tag{8}$$

This is equivalent to minimizing $L_2(V)$ as (6), except that the L2 constraint is replaced by a quadratic penalty term. When only samples are available, the expectation in (8) is replaced by the empirical version. If the optimization domain of $f$ is *unconstrained*, solving the empirical (8) reduces to the empirical L2 loss (1), which yields inconsistent estimation. Therefore, existing works propose to further constrain the optimization of $f$ in (8) to either RKHS (Dai et al., 2017) or neural networks (Dai et al., 2018b), and hence derive a minimax strategy for learning $V$. Unfortunately, this is substantially more expensive than our method due to the cost of updating another neural network $f$ jointly; the minimax procedure may also make the training less stable and more difficult to converge in practice.

## 3.3 Connection to Temporal Difference (TD) Methods

We now instantiate our algorithm in the *tabular* and *linear* cases to gain further insights. Interestingly, we show that our loss coincides with previous work, and as a result leads to the same value function as several classic algorithms. Hence, the approach developed here may be considered as their strict extensions to the much more general nonlinear function approximation classes.

Again, let $\mathcal{D}$ be a set of $n$ transitions sampled from distribution $\mu$, and linear approximation be used: $V_\theta(s) = \theta^{\mathrm{T}}\phi(s)$, where $\phi : S \to \mathbb{R}^d$ is a feature function, and $\theta \in \mathbb{R}^d$ is the parameter to be learned. The TD solution, $\hat{\theta}_{\mathrm{TD}}$, for either on- and off-policy cases, can be found by various algorithms (e.g., Sutton, 1988; Boyan, 1999; Sutton et al., 2009; Dann et al., 2014), and its theoretical properties have been extensively studied (e.g., Tsitsiklis & Van Roy, 1997; Lazaric et al., 2012).

**Corollary 3.5.** *When using a linear kernel of form $k(s, \bar{s}) = \phi(s)^{\mathrm{T}}\phi(\bar{s})$, minimizing the kernel objective* (4) *gives the TD solution $\hat{\theta}_{\mathrm{TD}}$.*

**Remark**    The result follows from the observation that our loss becomes the Norm of the Expected TD Update (NEU) in the linear case (Dann et al., 2014), whose minimizer coincides with $\hat{\theta}_{\mathrm{TD}}$. Moreover, in finite-state MDPs, the corollary includes tabular TD as a special case, by using a one-hot vector (indicator basis) to represent states. In this case, the TD solution coincides with that of a model-based approach (Parr et al., 2008) known as *certainty equivalence* (Kumar & Varaiya, 1986). It follows that our algorithm includes certainty equivalence as a special case in finite-state problems.

# 4  Kernel Loss for Policy Optimization

There are different ways to extend our approach to policy optimization. One is to use the kernel loss (3) inside an existing algorithm, as an alternative to RG or TD to learn $V^\pi(s)$. For example, our loss fits naturally into an actor-critic algorithm, where we replace the critic update (often implemented by TD($\lambda$) or its variant) with our method, and the actor updating part remains unchanged. Another, more general way is to design a kernelized loss for $V(s)$ and policy $\pi(a|s)$ jointly, so that the policy optimization can be solved using a single optimization procedure. Here, we take the first approach, leveraging our method to improve the critic update step in Trust-PCL (Nachum et al., 2018).

Trust-PCL is based on a temporal/path consistency condition resulting from policy smoothing (Nachum et al., 2017). We start with the smoothed Bellman operator, defined by

$$\mathcal{B}_\lambda V(s) = \max_{\pi(\cdot|s) \in \mathcal{P}_\mathcal{A}} \mathbb{E}_\pi[R(s,a) + \gamma V(s') + \lambda H(\pi \mid s) \mid s], \tag{9}$$

where $\mathcal{P}_\mathcal{A}$ is the set of distributions over action space $\mathcal{A}$; the conditional expectation $\mathbb{E}_\pi[\cdot|s]$ denotes $a \sim \pi(\cdot|s)$, and $\lambda > 0$ is a smoothing parameter; $H$ is a state-dependent entropy term: $H(\pi \mid s) := -\sum_{a \in \mathcal{A}} \pi(a|s) \log \pi(a|s)$. Intuitively, $\mathcal{B}_\lambda$ is a smoothed approximation of $\mathcal{B}$. It is known that $\mathcal{B}_\lambda$ is a $\gamma$-contraction (Fox et al., 2016), so has a unique fixed point $V_\lambda^*$. Furthermore, with $\lambda = 0$ we recover the standard Bellman operator, and $\lambda$ smoothly controls $\|V_\lambda^* - V^*\|_\infty$ (Dai et al., 2018b).

The entropy regularization above implies the following path consistency condition. Let $\pi_\lambda^*$ be an optimal policy in (9) for $\mathcal{B}_\lambda$, which yields $V_\lambda^*$. Then, $(V, \pi) = (V_\lambda^*, \pi_\lambda^*)$ uniquely solves

$$\forall (s, a) \in \mathcal{S} \times \mathcal{A} : \quad V(s) = R(s, a) + \gamma \mathbb{E}_{s'|s,a}[V(s')] - \lambda \log \pi(a|s).$$

This property inspires a natural extension of the kernel loss (3) to the controlled case:

$$L_K(V) = \mathbb{E}_{s,\bar{s} \sim \mu, a \sim \pi(\cdot|s), \bar{a} \sim \pi(\cdot|\bar{s})}[K([s,a],[\bar{s},\bar{a}]) \cdot \mathcal{R}_{\pi,\lambda}V(s,a) \cdot \mathcal{R}_{\pi,\lambda}V(\bar{s},\bar{a})],$$

where $\mathcal{R}_{\pi,\lambda}V(s,a)$ is given by

$$\mathcal{R}_{\pi,\lambda}V(s,a) = R(s,a) + \gamma \mathbb{E}_{s'|s,a}[V(s')] - \lambda \log \pi(a|s) - V(s).$$

Given a set of transitions $\mathcal{D} = \{(s_i, a_i, r_i, s_i')\}_{1 \le i \le n}$, the objective can be estimated by

$$\hat{L}_K(V_\theta) = \frac{1}{n^2} \sum_{1 \le i,j \le n} [K([s_i, a_i], [s_j, a_j])\hat{\mathcal{R}}_i\hat{\mathcal{R}}_j],$$

with

$$\hat{\mathcal{R}}_i = r_i + \gamma V_\theta(s_i') - \lambda \log \pi_\theta(a_i|s_i) - V_\theta(s_i).$$

The U-statistics version and multi-step bootstraps can be similarly obtained (Nachum et al., 2017).

## 5 Related Work

In this work, we studied value function learning, one of the most-studied and fundamental problems in reinforcement learning. The dominant approach is based on fixed-point iterations (Bertsekas & Tsitsiklis, 1996; Szepesvári, 2010; Sutton & Barto, 2018), which can risk instability and even divergence when function approximation is used, as discussed in the introduction.

Our approach exemplifies more recent efforts that aim to improve stability of value function learning by reformulating it as an optimization problem. Our key innovation is the use of a kernel method to estimate the squared Bellman error, which is otherwise hard to estimate directly from samples, thus avoids the double-sample issue unaddressed by prior algorithms like residual gradient (Baird, 1995) and PCL (Nachum et al., 2017, 2018). As a result, our algorithm is *consistent*: it finds the true value function with enough data, using sufficiently expressive function approximation classes. Furthermore, the solution found by our algorithm minimizes the projected Bellman error, as in prior works when specialized to the same settings (Sutton et al., 2009; Maei et al., 2010; Liu et al., 2015; Macua et al., 2015). However, our algorithm is more general: it allows us to use nonlinear value function classes and can be naturally implemented for policy optimization. Compared to nonlinear GTD2/TDC (Maei et al., 2009), our method is simpler (without having to do a local linear expansion) and empirically more effective (as demonstrated in the next section).

As discussed in Section 3, our approach is related to the recently proposed SBEED algorithm (Dai et al., 2018b) which shares many advantages with this work. However, SBEED requires solving a minimax problem that can be rather challenging in practice. In contrast, our algorithm only needs to solve a minimization problem, for which a wide range of powerful methods exist (e.g., Bertsekas, 2016). Note that there exist other saddle-point formulations for RL, which is derived from the linear program of MDPs (Wang, 2017; Chen et al., 2018; Dai et al., 2018a). The connection and comparison between these formulations would be interesting directions to investigate.

Our work is also related to a line of interesting work on Bellman residual minimization (BRM) based on nested optimization (Antos et al., 2008; Farahmand et al., 2008, 2016; Hoffman et al., 2011; Chen & Jiang, 2019). They formulate the value function as the solution to a coupled optimization problem, where both the inner and outer optimization are over the same function space. While their inner optimization plays a similar role as our use of RKHS in the kernel loss definition, our loss is derived from a different way, and decouples the representations used in inner and outer optimizations. Furthermore, the nested optimization formulation also involves solving a minimax problem (similar to SBEED), while our approach is much simpler as it only requires solving a minimization problem.

Finally, the kernel method has been widely used in machine learning (e.g., Schölkopf & Smola, 2001; Muandet et al., 2017). In RL, authors have used kernels either to smooth the estimates of transition probabilities and rewards (Ormoneit & Sen, 2002), or to represent the value function (e.g., Xu et al., 2005, 2007; Taylor & Parr, 2009). Our method differs from these works in that we leverage kernels for *designing proper loss functions* to address the double-sampling problem, while putting no constraints on which approximation classes to represent the value function. Our approach is thus expected to be more flexible and scalable in practice, allowing the value function to lie in flexible function classes like neural networks.

## 6 Experiments

We compare our method (labelled "K-loss" in all experiments) with several representative baselines in both classic examples and popular benchmark problems, for both policy evaluation and optimization.

### 6.1 Modified Example of Tsitsiklis & Van Roy

Fig. 1 (a) shows a modified problem of the classic example by Tsitsiklis & Van Roy (1997), by making transitions stochastic.[1] It consists of $5$ states, including $4$ nonterminal (circles) and $1$ terminal states (square), and $1$ action. The arrows represent transitions between states. The value function estimate is linear in the weight $\boldsymbol{w} = [w_1, w_2, w_3]$: for example, the leftmost and bottom-right states' values are $w_1$ and $2w_3$, respectively. Furthermore, we set $\gamma = 1$, so $V(s)$ is exact with the optimal

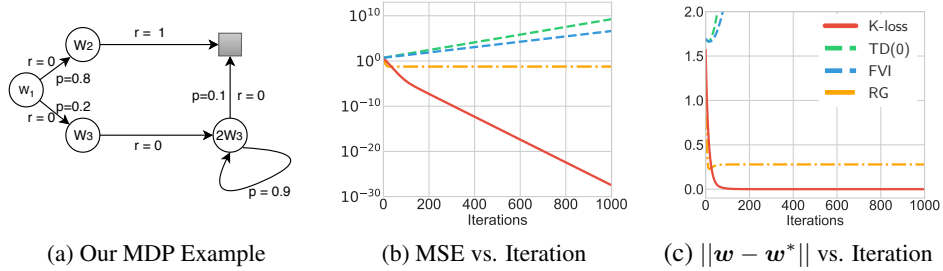

(a) Our MDP Example      (b) MSE vs. Iteration      (c) $||\boldsymbol{w} - \boldsymbol{w}^*||$ vs. Iteration

Figure 1: Modified example of Tsitsiklis & Van Roy (1997).

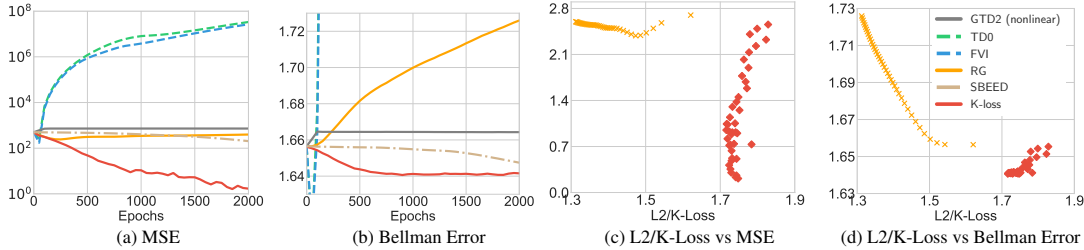

(a) MSE     (b) Bellman Error     (c) L2/K-Loss vs MSE     (d) L2/K-Loss vs Bellman Error

Figure 2: Results on Puddle World.

weight $\boldsymbol{w}^* = [0.8, 1.0, 0]$. In the experiment, we randomly collect $2\,000$ transition tuples for training. We use a linear kernel in our method, so that it will find the TD solution (Corollary 3.5).

Fig. 1 (b&c) show the learning curves of mean squared error ($||V - V^*||^2$) and weight error ($||\boldsymbol{w} - \boldsymbol{w}^*||$) of different algorithms over iterations. Results are consistent with theory: our method converges to the true weight $\boldsymbol{w}^*$, while both FVI and TD(0) diverge, and RG converges to a wrong solution.

## 6.2 Policy Evaluation with Neural Networks

While popular in recent RL literature, neural networks are known to be unstable for a long time. Here, we revisit the classic divergence example of Puddle World (Boyan & Moore, 1995), and demonstrate the stability of our method. Experimental details are found in Appendix B.2.

Fig. 2 summarizes the result using a neural network as value function for two metrics: $||V - V^*||_2^2$ and $||\mathcal{B}V - V||_2^2$, both evaluated on the training transitions. First, as shown in (a-b), our method works well while residual gradient converged to inferior solutions. In contrast, FVI and TD(0) exhibit unstable/oscilating behavior, and can even diverge, which is consistent with past findings (Boyan & Moore, 1995). In addition, non-linear GTD2 (Maei et al., 2009) and SBEED (Dai et al., 2017, 2018b), which do not find a better solution than our method in terms of MSE.

Second, Fig. 2 (c&d) show the correlation between MSE, empirical Bellman error of the value function estimation and an algorithm's training objective respectively. Our kernel loss appears to be a good proxy for learning the value function, for both MSE and Bellman error. In contrast, the L2 loss (used by residual gradient) does not correlate well, which also explains why residual gradient has been observed not to work well empirically.

Fig. 3 shows more results on value function learning on CartPole and Mountain Car, which again demonstrate that our method performs better than other methods in general.

## 6.3 Policy Optimization

To demonstrate the use of our method in policy optimization, we combine it with Trust-PCL, and compare with variants of Trust-PCL combined with FVI, TD(0) and RG. To fairly evaluate the performance of all these four methods, we use Trust-PCL (Nachum et al., 2018) framework and the public code for our experiments. We only modify the training of $V_\theta(s)$ for each of the method and keep rest same as original release. Experimental details can be found in Appendix B.3.1.

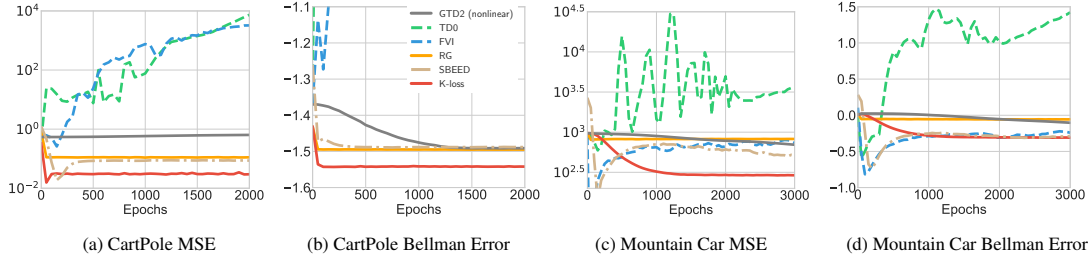

| (a) CartPole MSE | (b) CartPole Bellman Error | (c) Mountain Car MSE | (d) Mountain Car Bellman Error |

Figure 3: Policy evaluation results on CartPole and Mountain Car.

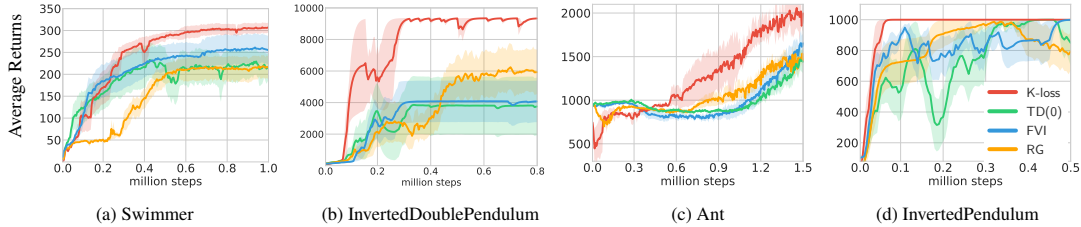

| (a) Swimmer | (b) InvertedDoublePendulum | (c) Ant | (d) InvertedPendulum |

Figure 4: Results of various variants of Trust-PCL on Mujoco Benchmark.

We evaluate the performance of these four methods on Mujoco benchmark and report the best performance of these four methods in Figure 4 (averaged on five different random seeds). K-loss consistently outperforms all the other methods, learning better policy with fewer data. Note that we only modify the update of value functions inside Trust-PCL, which can be implemented relatively easily. We expect that we can improve many other algorithms in similar ways by improving the value function using our kernel loss.

## 7 Conclusion

This paper studies the fundamental problem of solving Bellman equations with parametric value functions. A novel kernel loss is proposed, which is easy to be estimated and optimized using sampled transitions. Empirical results show that, compared to prior algorithms, our method is convergent, produces more accurate value functions, and can be easily adapted for policy optimization.

These promising results open the door to many interesting directions for future work. An important question is finite-sample analysis, quantifying how fast the minimizer of the empirical kernel loss converges to the true minimizer of the population loss, when data is *not* i.i.d. Another is to extend the loss to the online setting, where data arrives in a stream and the learner cannot store all previous data. Such an online version may provide computational benefits in certain applications. Finally, it may be possible to quantify uncertainty in the value function estimate, and use this uncertainty information to guide efficient exploration.

## Acknowledgment

This work is supported in part by NSF CRII 1830161 and NSF CAREER 1846421. We would like to acknowledge Google Cloud and and Amazon Web Services (AWS) for their support. We also thank an anonymous reviewer and Bo Dai for helpful suggestions on related work that improved the paper.

## Footnotes

[1]Recall that the double-sample issue exists only in stochastic problems, so the modification is necessary to make the comparison to residual gradient meaningful.

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
