[Supplementary Material]

# Appendix

## A  Proofs for Section 3

### A.1  Proof of Theorem 3.1

The assertion that $L_K(V) \geq 0$ for all $V$ is immediate from definition. For the second part, we have

$$
\begin{aligned}
L_K(V) = 0 &\iff \|\mathcal{R}_\pi V\|_{K,\mu} = 0 \\
&\iff \|\mathcal{R}_\pi V \cdot \mu\|_K = 0 \\
&\iff \mathcal{R}_\pi V(s)\mu(s) = 0, \quad \forall s \in \mathcal{S} \qquad \text{(since } K \text{ is an is ISPD kernel)} \\
&\iff \mathcal{R}_\pi V(s) = 0 \quad \forall s \in \mathcal{S} \\
&\iff V = V^\pi.
\end{aligned}
$$

### A.2  Proof of Theorem 3.2

Define $\delta = V - V^\pi$ to be the value function error. Furthermore, let $\mathcal{I}$ be the identity operator $(\mathcal{I}V = V)$, and

$$
\mathcal{P}_\pi V(s) := \mathbb{E}_{a \sim \pi(\cdot|s), s' \sim P(\cdot|s,a)}[\gamma V(s') \mid s]
$$

the state-transition part of Bellman operator without the local reward term $R(s,a)$.

Note that $\mathcal{R}_\pi V^\pi = \mathcal{B}_\pi V^\pi - V^\pi = 0$ by the Bellman equation, so

$$
\mathcal{R}_\pi V = \mathcal{R}_\pi V - \mathcal{R}_\pi V^\pi = (\mathcal{P}_\pi V - V) - (\mathcal{P}_\pi V^\pi - V^\pi) = (\mathcal{P}_\pi - \mathcal{I})(V - V^\pi) = (\mathcal{P}_\pi - \mathcal{I})\delta.
$$

Therefore,

$$
\begin{aligned}
L_K(V) &= \mathbb{E}_\mu[\mathcal{R}_\pi V(s) \cdot \mathcal{R}_\pi V(\bar{s}) \cdot K(s,\bar{s})] \\
&= \mathbb{E}_\mu[(\mathcal{I} - \mathcal{P}_\pi)\delta(s) \cdot (\mathcal{I} - \mathcal{P}_\pi)\delta(\bar{s}) \cdot K(s,\bar{s})] \\
&= \mathbb{E}_{(s,s'),(\bar{s},\bar{s}') \sim d_{\pi,\mu}}[(\delta(s) - \gamma\delta(s')) \cdot (\delta(\bar{s}) - \gamma\delta(\bar{s}')) \cdot K(s,\bar{s})],
\end{aligned}
$$

where $\mathbb{E}_{d_{\pi,\mu}}[\cdot]$ denotes the expectation under the joint distribution

$$
d_{\pi,\mu}(s, s') := \mu(s) \sum_{a \in \mathcal{A}} \pi(a|s)P(s'|s,a).
$$

Expanding the quadratic form above, we have

$$
\begin{aligned}
&L_K(V) \\
&= \mathbb{E}_{d_{\pi,\mu}}[(\delta(s)K(s,\bar{s})\delta(\bar{s}) - \gamma\delta(s')\delta(\bar{s})K(s,\bar{s}) - \gamma\delta(\bar{s}')\delta(s)K(s,\bar{s}) + \gamma^2\delta(s')\delta(\bar{s}')K(s,\bar{s})] \\
&= \mathbb{E}_\mu[\delta(s')K^*(s',\bar{s}')\delta(\bar{s}')],
\end{aligned}
$$

where $K^*(s',\bar{s}')$ is as defined in the theorem statement:

$$
K^*(s',\bar{s}') = \mathbb{E}_{d^*_{\pi,\mu}}\left[K(s',\bar{s}') - \gamma(K(s',\bar{s}) + K(s,\bar{s}')) + \gamma^2 K(s,\bar{s}) \mid (s',\bar{s}')\right]
$$

with the expectation w.r.t. the following "backward" conditional probability

$$
d^*_{\pi,\mu}(s \mid s') := \frac{\sum_{a \in \mathcal{A}} \pi(a|s)P(s'|s,a)\mu(s)}{\mu(s')},
$$

which can be heuristically viewed as the distribution of state $s$ conditioning on observing its next state $s'$ when following $d_{\pi,\mu}(s, s')$.

### A.3  Proof of Proposition 3.3

Using the eigen-decomposition (5), we have

$$
\begin{aligned}
L_K(V) &= \mathbb{E}_\mu[\mathcal{R}_\pi V(s)K(s,\bar{s})\mathcal{R}_\pi V(\bar{s})] \\
&= \mathbb{E}_\mu[\mathcal{R}_\pi V(s) \sum_{i=1}^\infty \lambda_i e_i(s)e_i(\bar{s})\mathcal{R}_\pi V(\bar{s})] \\
&= \sum_{i=1}^\infty \lambda_i \left(\mathbb{E}_\mu[\mathcal{R}_\pi V(s)e_i(s)]\right)^2.
\end{aligned}
$$

The decomposition of $L_2(V)$ follows directly from Parseval's identity.

## A.4   Proof of Proposition 3.4

The reproducing property of RKHS implies $f(s) = \langle f, K(s, \cdot)\rangle_{\mathcal{H}_K}$ for any $f \in \mathcal{H}_K$. Therefore,

$$
\begin{aligned}
\mathbb{E}_\mu[\mathcal{R}_\pi V(s) f(s)] &= \mathbb{E}_\mu[\mathcal{R}_\pi V(s) \langle f, K(s, \cdot)\rangle_{\mathcal{H}_K}]\\
&= \langle f, \mathbb{E}_\mu[\mathcal{R}_\pi V(s) K(s, \cdot)]\rangle_{\mathcal{H}_K}\\
&= \langle f, f^*\rangle_{\mathcal{H}_K}.
\end{aligned}
$$

where we have defined $f^*(\cdot) := \mathbb{E}_\mu[\mathcal{R}_\pi V(s) K(s, \cdot)]$. Maximizing $\langle f, f^*\rangle$ subject to $\|f\|_{\mathcal{H}_K} := \sqrt{\langle f, f\rangle_{\mathcal{H}_K}} \leq 1$ yields that $f = f^*/\|f^*\|_{\mathcal{H}_K}$. Therefore,

$$
\max_{f \in \mathcal{H}_K: \|f\|_{\mathcal{H}_K} \leq 1} (\mathbb{E}_s[\mathcal{R}_\pi V(s) f(s)])^2 = (\langle \frac{f^*}{\|f^*\|_{\mathcal{H}_K}}, f^*\rangle_{\mathcal{H}_K})^2 = \|f^*\|^2_{\mathcal{H}_K}.
$$

Further, we can show that

$$
\begin{aligned}
\|f^*\|^2_{\mathcal{H}_K} &= \langle f^*, f^*\rangle_{\mathcal{H}_K}\\
&= \langle \mathbb{E}_\mu[\mathcal{R}_\pi V(s) K(s, \cdot)], \ \mathbb{E}_\mu[\mathcal{R}_\pi V(\bar{s}) K(\bar{s}, \cdot)]\rangle_{\mathcal{H}_K}\\
&= \mathbb{E}_\mu[\mathcal{R}_\pi V(s) K(s, \bar{s}) \mathcal{R}_\pi V(\bar{s})],
\end{aligned}
$$

where the last step follows from the reproducing property, $K(s, \bar{s}) = \langle K(s, \cdot), K(\bar{s}, \cdot)\rangle_{\mathcal{H}_K}$. This completes the proof, by definition of $L_K(V)$.

## A.5   Proof of Corollary 3.5

Under the conditions of the corollary, the kernel loss becomes the Norm of the Expected TD Update (NEU), whose minimizer coincides with the TD solution (Dann et al., 2014). For completeness, we provide a self-contained proof.

Since we are estimating the value function of a fixed policy, we ignore the actions, and the set of transitions is $\mathcal{D} = \{(s_i, r_i, s_i')\}_{1 \leq i \leq n}$. Define the following vector/matrices:

$$
\begin{aligned}
r &= [r_1; \ r_2, \cdots; \ r_n] \in \mathbb{R}^{n \times 1},\\
X &= [\phi(s_1); \ \phi(s_2); \ \ldots; \ \phi(s_n)] \in \mathbb{R}^{n \times d},\\
X' &= [\phi(s_1'); \ \phi(s_2'); \ \ldots; \ \phi(s_n')] \in \mathbb{R}^{n \times d},
\end{aligned}
$$

and $Z = X - \gamma X'$, where $d$ is the feature dimension. Then, the TD solution is given by

$$
\hat{\theta}_{\mathrm{TD}} = (X^\mathrm{T} Z)^{-1} X^\mathrm{T} r.
$$

Note that the above includes both the on-policy case as well as the off-policy case as in many previous algorithms with linear value function approximation (Dann et al., 2014), where the difference is in whether $s_i$ is sampled from the state occupation distribution of the target policy or not.

Define $\delta \in \mathbb{R}^{n \times 1}$ to be the TD error vector; that is, $\delta = r - Z\theta$, where $\delta_i = r_i + \gamma V(s_i') - V(s_i) = r_i + \theta^\mathrm{T}(\gamma\phi(s_i') - \phi(s_i))$. With a linear kernel, our objective function becomes:

$$
\ell(\theta) = \frac{1}{n^2}\sum_{i,j}\delta_i K(s_i, s_j)\delta_j = \frac{1}{n^2}\delta^\mathrm{T} X X^\mathrm{T}\delta = \frac{1}{n^2}(r - Z\theta)^\mathrm{T} X X^\mathrm{T}(r - Z\theta).
$$

Its gradient is given by

$$
\nabla\ell = \frac{2}{n^2}(Z^\mathrm{T} X X^\mathrm{T} Z\theta - Z^\mathrm{T} X X^\mathrm{T} r).
$$

Letting $\nabla\ell = 0$ gives the solution obtained by minimizing our kernel loss:[2]

$$
\hat{\theta}_{\mathrm{KBE}} = (Z^\mathrm{T} X X^\mathrm{T} Z)^{-1} Z^\mathrm{T} X X^\mathrm{T} r.
$$

Therefore,

$$
\begin{aligned}
\hat{\theta}_{\mathrm{KBE}} - \hat{\theta}_{\mathrm{TD}} &= \left((Z^\mathrm{T} X X^\mathrm{T} Z)^{-1} Z^\mathrm{T} X - (X^\mathrm{T} Z)^{-1}\right) X^\mathrm{T} r\\
&= \left((Z^\mathrm{T} X X^\mathrm{T} Z)^{-1} Z^\mathrm{T} X (X^\mathrm{T} Z) - I\right)(X^\mathrm{T} Z)^{-1} X^\mathrm{T} r\\
&= (I - I)(X^\mathrm{T} Z)^{-1} X^\mathrm{T} r = 0.
\end{aligned}
$$

# B Experiment Details

## B.1 Kernel Loss Estimation with Batch Samples

Given a set of empirical data $\mathcal{D} = \{(s_i, a_i, r_i, s_i')\}_{1 \leq i \leq n}$, where $n$ is large such that we need to use a subset samples $\mathcal{B} = \{(S_i, A_i, R_i, S_i')\}_{1 \leq i \leq m}$ drawn from $\mathcal{D}$ to estimate the empirical kernel loss. One way to estimate $L_K$ using the subset $\mathcal{B}$ is *U-statistics*,

$$\hat{L}_{KU}(V_\theta) := \frac{1}{m(m-1)} \sum_{1 \leq i \neq j \leq m} K(S_i, S_j) \cdot \hat{\mathcal{R}}_\pi V_\theta(S_i) \cdot \hat{\mathcal{R}}_\pi V_\theta(S_j) \,.$$

Similarly, we can use the *V-statistics* to estimate $L_K$ given the subset $\mathcal{B}$:

$$\hat{L}_{KV}(V_\theta) := \frac{1}{mn} \left( \left( \sum_{1 \leq i \leq m} K(S_i, S_i) \cdot \hat{\mathcal{R}}_\pi V_\theta(S_i) \cdot \hat{\mathcal{R}}_\pi V_\theta(S_i) \right) \right.$$
$$\left. + \frac{n-1}{m-1} \left( \sum_{1 \leq i \neq j \leq m} K(S_i, S_j) \cdot \hat{\mathcal{R}}_\pi V_\theta(S_i) \cdot \hat{\mathcal{R}}_\pi V_\theta(S_j) \right) \right) \,.$$

In our experiments, we observe that V-statistics works slightly better than U-statistics, and we use an mixed combination of these two to achieve better performance: $\alpha \hat{L}_{KV}(V_\theta) + (1-\alpha)\hat{L}_{KU}(V_\theta)$, where $\alpha$ is a coefficient which can be tunned.

## B.2 Policy Evaluation

We compare our method with representative policy evaluation methods including TD(0), FVI, RG, nonlinear GTD2 (Maei et al., 2009) and SBEED (Dai et al., 2017, 2018b) on three different stochastic environments: Puddle World, CartPole and Mountain Car. Followings are the detail of the policy evaluation experiments.

**Network Structure** We parameterize the value function $V_\theta(s)$ using a fully connected neural network with one hidden layer of 80 units, using `relu` as activation function. For test function $f(s)$ in SBEED, we use a small neural network with 10 hidden units and `relu` as activation function.

**Data Collection** For each environment, we randomly collect 5000 independent transition tuples with states uniformly drawn from state space using a policy $\pi$ learned by policy optimization, for which we want to learn the value function $V^\pi(s)$.

**Estimating the true value function** $V^\pi(s)$ To evaluate and compare all methods, we approximate the true value function by finely discretizing the state space and then applying tabular value iteration on the discretized MDP. Specifically, we discretize the state space into $25 \times 25$ grid for Puddle World, $20 \times 25$ discrete states for CartPole, and $30 \times 25$ discrete states for Mountain Car.

**Training Details** For each environment and each policy evaluation method, we train the value function $V_\theta(s)$ on the collected 5000 transition tuples for 2000 epochs (3000 for Mountain Car), with a batch size $n = 150$ in each epoch using Adam optimizer. We search the learning rate in $\{0.003, 0.001, 0.0003\}$ for all methods and report the best result averaging over 10 trials using different random seeds. For our method, we use a Gaussian RBF kernel $K(s_i, s_j) = \exp\left(-\|s_i - s_j\|_2^2 / h^2\right)$ and take the bandwith to be $h = 0.5$. For FVI, we update the target network at the end of each epoch training. For SBEED, we perform 10 times gradient ascent updates on the test function $f(s)$ and 1 gradient descent update on $V_\theta(s)$ at each iteration. We fix the discount factor to $\gamma = 0.98$ for all environments and policy evaluation methods.

## B.3 Policy Optimization

In this section we describe in detail the experimental setup for policy optimization regarding implementation and hyper-parameter search. The code of Trust-PCL is available at github.[3] Algorithm 1 describes details in pseudocode, where the the main change compared to Trust-PCL is highlighted. Note that as in previous work, we use the $d$-step version of Bellman operator, an immediate extension to the $d = 1$ case described in the main text.

**Algorithm 1** K-Loss for PCL

---

**Input:** rollout step $d$, batch size $B$, coefficient $\lambda$, $\tau, \alpha$.

Initialize $V_\theta(s)$, $\pi_\phi(a|s)$, and empty replay buffer $RB(\beta)$. Set $\tilde{\phi} = \phi$.

**repeat**

    *// Collecting Samples*
    Sample $P$ steps $s_{t:t+P} \sim \pi_\phi$ on ENV.
    Insert $s_{t:t+P}$ to $RB(\beta)$.

    *// Train*
    Sample batch $\{s_{t:t+d}^{(k)}, a_{t:t+d}^{(k)}, r_{t:t+d}^{(k)}\}_{k=1}^{B}$ from $RB(\beta)$ to contain a total of $Q$ transitions ($B \approx Q/d$).
    $\Delta\theta = \alpha\nabla_\theta \hat{L}_{KV}(V_\theta) + (1-\alpha)\nabla_\theta \hat{L}_{KU}(V_\theta),$
    $\Delta\phi = -\frac{1}{B}\sum_{1\le i \le B}[\hat{\mathcal{R}}_i \sum_{t=0}^{d-1}\nabla_\phi \log \pi_\phi(a_{t+i}|s_{t+i})]$, where

$$\hat{\mathcal{R}}_i = -V_\theta(s_i) + \gamma^d V_\theta(s_{i+d}) + \sum_{t=0}^{d-1}\gamma^t(r_{i+t} - (\lambda+\tau)\log\pi_\phi(a_{t+i}|s_{t+1}) + \tau\log\pi_{\tilde{\phi}}(a_{t+i}|s_{t+1})).$$

    Update $\theta$ and $\phi$ using ADAM with $\Delta\theta, \Delta\phi$.

    *// Update auxiliary variables*
    Update $\tilde{\phi} = \alpha\tilde{\phi} + (1-\alpha)\phi$.

**until** Convergence

---

Figure 5: More results of various variants of Trust PCL on Mujoco Benchmark (on top of Figure 4).

## B.3.1 Network Architectures

We use fully-connected feed-forward neural network to represent both policy and value network. The policy $\pi_\theta$ is represented by a neural network with $64 \times 64$ hidden layers with `tanh` activations. At each time step $t$, the next action $a_t$ is sampled randomly from a Gaussian distribution $\mathcal{N}(\mu_\theta(s_t), \sigma_\theta)$. The value network $V_\theta(s)$ is represented by a neural network with $64 \times 64$ hidden layers with `tanh` activations. At each time step $t$, the network is given the observation $s_t$ and it produces a single scalar output value. All methods share the same policy and value network architectures.

## B.3.2 Training Details

We average over the best 5 of 6 randomly seeded training runs and evaluate each method using the mean $\mu_\theta(s)$ of the diagonal Gaussian policy $\pi_\theta$. Since Trust-PCL is off-policy, we collect experience and train on batches of experience sampled from the replay buffer. At each training iteration, we will first sample $T = 10$ timestep samples and add them to the replay buffer, then both the policy and value parameters are updated in a single gradient step using the Adam optimizer with a proper learning rate searched, using a minibatch randomly sampled from replay buffer. For Trust-PCL using FVI updating $V_\theta(s)$, which requires a target network to estimate the final state for each path, we use an exponentially moving average, with a smoothing constant $\tau = 0.99$, to update the target value network weights as common in the prior work (Mnih et al., 2015). For Trust-PCL using TD(0), we will directly use current value network $V_\theta(s)$ to estimate the final states except we do not perform gradient update for the final states. For Trsut-PCL using RG and K-loss, which has an objective loss, we will directly perform gradient descent to optimize both policy and value parameters.

### B.3.3 Hyperparameter Search

We follow the same hyperparameter search procedure in Nachum et al. (2018) for FVI, TD(0) and RG based Trust-PCL.[4] We search the maximum divergence $\epsilon$ between $\pi_\theta$ and $\pi_{\hat{\theta}}$ among $\in$ $\{0.001, 0.0005, 0.002\}$, and parameter learning rate in $\{0.001, 0.0003, 0.0001\}$, and the rollout length $d \in \{1, 5, 10\}$. We also searched with the entropy coefficient $\lambda$, either keeping it at a constant $0$ (thus, no exploration) or decaying it from $0.1$ to $0.0$ by a smoothed exponential rate of $0.1$ every $2500$ training iterations. For each hyper-parameter setting, we average best $5$ of $6$ seeds and report the best performance for these methods.

For our proposed K-loss, we also search the maximum divergence $\epsilon$ but keep the learning rate as $0.001$. Additionally, for K-loss we use a Gaussian RBF kernel $K([s_i, a_i], [s_j, a_j]) = \exp\left(-(\|s_i - s_j\|_2^2 + \|a_i - a_j\|_2^2)/h\right)$, and take the bandwidth to be $h = (\alpha \times \text{med})^2$, where we search $\alpha \in \{0.1, 0.01, (1/\sqrt{\log B})\}$, and $B = 64$ is the gradient update batch size. We fix the discount to $\gamma = 0.995$ for all environments and batch size $B = 64$ for each training iteration.

## Footnotes

[2]For simplicity, we assume all involved matrices of size $d \times d$ are non-singular, as is typical in analyzing TD algorithms. Without this assumption, we may either add $L_2$-regularization to $X X^\mathrm{T}$ (Farahmand et al., 2008), for which the same equivalence between TD and ours can be proved, or show that the solutions lie in an affine space in $\mathbb{R}^d$ but the corresponding value functions are identical.

[3] https://github.com/tensorflow/models/tree/master/research/pcl_rl

[4]See README in `https://github.com/tensorflow/models/tree/master/research/pcl_rl`