[Reviews · NeurIPS 2019]

Reviewer 1



The paper proposes a new loss function for value function learning. This loss function is the Bellman error measured in a norm based on an ISPD kernel. The motivation is that this loss can be optimized from data without divergence (compared to projected Bellman equation), and can be estimated from data in a consistent way (compared to residual gradient). The method is shown to outperform various baselines when combined with PCL for policy optimization. This is a strong and well-written work, and I enjoyed reading it. The contribution is a fresh perspective on approximating value functions, and it is nice to see that this creative turn away from the literature (which is heavily based on the projected Bellman equation) leads to strong results, both theoretically and empirically. The empirical evaluation is convincing, and the example in Figure 1 clearly illustrates the benefit of the proposed approach. The 3.2 relates the loss function to an error in the value function. However, this error is measured w.r.t. a dual kernel that I found hard to interpret. Can the authors relate it to a more meaningful quantity? E.g., measuring error in L-p norms in very intuitive, and I wonder whether these results can be related in some way. One issue that wasn’t discussed was the computation time. For N data samples, each gradient step would require O(N^2) computations, which can be limiting for big-data regimes. I wonder if the authors can comment on this. Minor comments: 66: this notation is a bit strange - what if there are multiple transitions from a state? 66: the parametric V_theta is used before it’s introduced a bit later on.

Reviewer 2



Originality: The derivation of the loss function is original; the resulting loss function has some close similarities with the coupled formulation of LSTD, which should be discussed. Quality: The claims seem to be accurate (I briefly verified the proofs of Theorem 3.1, Proposition 3.3, Proposition 3.4; I did not verify Theorem 3.2 and Corollary 3.5). Clarity: The paper is well-written and clear. Significance: The addressed problem is important; the insights are also useful. SUMMARY: The paper addresses the problem of designing a new loss function for RL. Fixed-point iteration types of algorithms (i.e., value iteration and variants) are based on iterative application of the Bellman operator. Unfortunately they may not converge when we use function approximator to represent the value function and use off-policy data. One way to think about this is that VI is not optimizing any loss function. An alternative is to design a loss function, which can then be minimized by a gradient descent-like algorithm, whose minimizer is the value function. This is exactly what this paper does. The Bellman error is such a loss function. But its empirical version, which is minimized by the residual gradient algorithm (Baird, 1995) does not lead to a satisfactory solution. The reason is that the empirical Bellman error is a biased estimate of the Bellman error, so its minimizer does not always have desired property (unless the dynamics is deterministic). One solution is to use double sampling, which is not always possible or desirable. This paper takes an ingenious approach of defining a loss function as an RKHS norm of a the Bellman residual R_pi V = B_pi V - V (with B_pi being the Bellman operator, more commonly denoted by T^pi). That is, the (population version of) loss is L_K(V) = || R_pi V ||_{K, mu}^2, where mu is a distribution over states and K is the kernel. Under conditions on the choice of kernel and the positivity of the distribution mu, the minimizer of this loss function is indeed the value function (Theorem 3.1). One can use data to estimate L_K(V) and its gradient based on a V or U estimator (Eq. 4). It is claimed that these estimates are consistent and does not have the same issue as we have with the empirical Bellman error, so we do not need to perform double sampling. The paper provides some interpretation of the suggested loss function. One is that it is the weighted projected Bellman error (Proposition 3.3). The projection is onto the span of eigenfunctions of the kernel function, and the weights are their corresponding eigenvalues. The paper also provides a variational interpretation of the loss function (Proposition 3.4). The paper shows how the method can be modified to perform policy optimization. Through a series of empirical results, it shows that the method performs quite well. EVALUATION: This is an interesting and well-written paper, which provides a new insight about how we may design a loss function for RL. I have two main issues/concerns about this paper, which I describe below: 1) The connection to the nested optimization formulation of LSTD is not discussed. Even though the derivation of the loss function starts from a different perspective, Proposition 3.3 shows that the loss function is indeed minimizing the projected Bellman error. As mentioned earlier, the projection is onto the span of eigenfunctions of the kernel function, and the weights are the eigenvalues. We can compare this with the formulations provided in the following papers: - Antos, Szepesvari, and Munos, MLJ, 2008. - Farahmand, Ghavamzadeh, Szepesvari, Mannor, NIPS, 2008. - Farahmand, Ghavamzadeh, Szepesvari, Mannor, “Regularized Policy Iteration with Nonparametric Function Spaces,” JMLR, 2016. - Hoffman, Lazaric, Ghavamzadeh, Munos, “Regularized least squares temporal difference learning with nested l1 and l2 penalization,” LNCS, 2012. For example, in Farahmand et al., 2016, LSTD is formulated as two coupled optimization problems (Eq. 10), one of which is computing the projection of B_pi Q onto the function space F, and the other minimizes the distance between the projection B_pi Q and Q. Proposition 3.3 shows that L_K is essentially the projection of B_pi V - V onto the span of an RKHS F, weighted by the eigenvalues. These are not exactly the same because: - The weightings here are based on the eigenvalues. The other formulation does not prescribe any particular weighting, though the weighting can be possibly included. - Here the projection is onto an RKHS, whereas Farahmand et al. does not necessarily formulate the problem as an RKHS (even though they provide an RKHS-based solution too). - Here V is not specified to belong to the RKHS F and it can be from another function space. Also see the discussion about LSTD (around Eq. 13) in Antos et al, 2008. To summarize this point, I think the loss function here is derived from a different perspective, but it is not completely novel and has close relationship with the coupled optimization formulation of the prior work. This requires further discussion and clarification in the paper. I should say that Antos et al., 2008 and Farahmand et al., 2008 are both cited, but they are not referred to in this crucial aspect (Farahmand et al., 2008 is cited as a technical comment in a footnote in the supplementary material). 2) The properties of the empirical loss are not shown. Does Eq. (4) provide an unbiased estimate? There is a discussion on the consistency of U and V estimators, but nothing is proved. Please correct me if I am wrong: Even if the data is generated i.i.d., (4) is biased, but if we have i \neq j in it, it will be unbiased. If the data is not i.i.d., then its bias depends on the dependence between states, as discussed in the Remark right after. The amount of bias depends on the amount of dependence of the Markov chain. So in some sense, we have not completely solved the double sampling issue. Moreover, it is not shown how close/far the minimizer of hat{L}_K(V_theta) would be to the minimizer of L_K(V_theta). Are they close? There is a comment in the conclusion about including finite-sample analysis, which I suppose refers to this question. Addressing this question may not be crucial at this point though. I think it is important to at least show some basic properties of these estimators, because a distinctive point of the new loss compared to the empirical Bellman error (and residual gradient algorithm) is exactly the issue of bias and the avoidance of double sampling. Other Questions and Comments: - What is the interplay of the function space to which V belongs and the RKHS corresponding to K (let’s call it H). To be specific, there is a comment at L237 which states that as opposed to some previous kernel-based methods, the method does not put any constraints on the approximation classes for representing the value function. It is true that the method does not require V to belong to H. But according to Proposition 3.3, the loss function penalizes the Bellman residual of V only in the span of H. Of course, it might be possible that V doesn’t belong to H but R_pi V does (but if K is a universal kernel, this wouldn’t happen). What I am suspecting, and I would appreciate if the authors provide some comments, is that the choice of the value function space may not matter as much as the choice of K does. - This might be a bit subjective, but I believe that the paper by Ormoneit & Sen, 2002 is not really a kernel-based method in the same sense as RKHS ones. Its use of kernel is in the sense of smoothing kernel methods. For example, refer to the discussion at the beginning of Chapter 6 of Hastie, Tibshirani, and Friedman, The Elements of Statistical Learning (Second Edition). - What is the purpose of Theorem 3.2? It seems a bit dormant and out of place. It is not used anywhere else either as far as I can see. Typos: - L36: Extra space “convergent .” - L101: R_pi —> R_pi V

Reviewer 3



***Post Rebuttal*** I read the other reviews and the author feedback. The authors clarified my few concerns, thus, I confirm my positive score. The paper is written in good English and reads very well. I think that this work represents a valuable contribution for NeurIPS and I vote for the acceptance. The proposed objective function, although not unbiased, is consistent, differently from the squared TD error and it can be more easily estimated, avoiding the double sample approach. The analysis of the proposed objective is fairly deep. I liked the connection with the L2 loss and with TD methods. The experimental evaluation succeded in showing the advantages of the approach over state-of-the-art methods, although the presentation of the results can be improved. I also checked the proofs in the appendix and everything seems correct to me. Here are my detailed comments. - line 66: here the value function depends on the parameter \theta, but the parametric value function is introduced only at line 69 - Figures 1, 2, and 3: there are no labels on the y-axis - Plots are not very readable, especially when the paper is printed in greyscale. I suggest to use different line styles or to introduce markers to differentiate the curves. - Figure 1b: I think that a y scale up to 10^{-30} is not necessary, 10^{-10} would be sufficient. Shouldn't the 0 tick on the y-axis be 10^{0} = 1? - Figure 1c: although they diverge, the light blue and green curves are plotted for very few iterations - Figures 2c and 2d: I did not fully get the meaning of these plots. Can the authors clarify better? -line 277-278: "report the best performance of these four methods". What do the authors mean by "best performance"? Maybe, the best hyperparameter configuration? ***Typos*** - line 36: blank space before comma - line 264: Fig. 2(c&d) -> Fig. 2 (c&d) - line 273: TD0 -> TD(0) - line 279: bettere -> better - line 280 and caption of Figure 4: Trust PCL -> Trust-PCL - line 472 and 498: footnote should be after the full stop

[Author Response · NeurIPS 2019]

We thank all the reviewers for their time and valuable feedback. We will improve the draft carefully based on your
comments.

**Response to Reviewer 1**

*Intuitive interpretation of the dual kernel norm (Thm 3.2)?*    The purpose of Thm 3.2 is to clarify how the kernel
Bellman loss is related to the error $\delta = V - V_\pi$. This is achieved by "shifting" the Bellman operator on $V(s)$ to the
kernel, yielding an *adjoint operator* applied on the kernel (and hence the dual kernel). That said, the definition of the
dual kernel is rather technically motivated. We tend to think it as *"the kernel whose kernel norm equals our kernel*
*Bellman loss.*

The (dual) kernel norm is not directly related to $L_p$ norm. But if the maximum (resp. minimum) eigenvalue of the
kernel is positive, then the kernel norm can be upper (resp. lower) bounded by $L_2$ norm. We will include more intuitive
interpretation in the revision.

*Computation time*    While the exact gradient involves $O(n^2)$ computation, we can obtain stochastic gradients using
mini-batches, whose computation does not scale with $n$. This process is straightforward for U-statistics. For V-statistics,
we need to estimate its diagonal and off-diagonal parts separately to obtain an unbiased stochastic gradient. Details will
be added in the final version.

**Response to Reviewer 2**

*The connection to the nested optimization formulation of LSTD.*   We thank the reviewer for insightfully relating this
work to nested optimization formulations of LSTD and BRM algorithms. We agree that they are closely related and
will give a thorough discussion in the revision. But our approach is derived from a very different perspective and is both
simpler and more practical for general nonlinear function classes, and differs in other ways including (1) the primal and
dual spaces can be specified independently; (2) we specifically project the Bellman residual onto RKHS that results
in a closed form solution; (3) algorithmically our method solves a single optimization problem without the two-step
procedure in previous work; among others.

Meanwhile, we believe that we can develop results similar to our Corollary 3.3 to explicitly clarify the concrete relation
between LSTD and kernel loss in some special cases. We will discuss this extensively in the revision.

*The properties of the empirical loss are not shown.*   We agree with the reviewer's comments on the issue of biasedness
(see also our remark in L116–123). We do want to emphasize the difference between unbiasedness and consistency
here. Even though our estimator is biased with non-iid data (even by using U-statistics), we still yield a consistent
estimator under standard regularity conditions of the Markov chain; this is much better than the squared TD error used
by residual gradient, which is inconsistent.

Establishing statistical properties of the estimator is an interesting problem, and can be achieved with the now-standard
methods. But the anslysis for the non-IID case is quite technical, and can be a distraction of this work's main focus.
Therefore, we prefer to study it in a separate work that focuses on statistical guarantees and uncertainty estimation.

*Interplay of the function space $V(s)$ belongs and the kernel corresponded RKHS.*   The role of the value function
space (denote it by $\mathcal{V}$) and kernel space $\mathcal{H}$ are similar to the generator and discriminator spaces in GAN (Goodfellow
et al. 14). They plays orthogonal roles, so it is not easy to say which is more important. But $\mathcal{V}$ and $\mathcal{H}$ do need to be
compatible with each other in that sense that $\mathcal{H}$ should be chosen to include $\{V - \mathcal{B}_\pi V \colon \forall V \in \mathcal{V}\}$, which holds when
$\mathcal{H}$ is universal.

*Clarification of the paper by Ormoneit & Sen, 2002.*   We agree with the reviewer that the kernel smoothing used in
"Kernel-Based Reinforcement Learning" is not the same as the more general kernel methods used in the paper and other
works. It is mentioned in the paper as a related work, and we will make the distinction explicit.

*The purpose of Thm 3.2.*   Thm 3.2 is meant to clarify how the kernel Bellman loss is related to the error $\delta = V - V_\pi$ of
the values functions (see also our response to Reviewer 1). We will consider to reform Theorem 3.2 into a "Dual kernel
interpretation" inside Section 3.2, so that it does not interrupt the presentation in Section 3.1.

**Response to Reviewer 3**

*The meaning of plots in Figure 2(c) & (d).*   The red dots in Fig2(c) are the (*MSE, K-loss*) pairs obtained in the trajectory
of our algorithm. The yellow dots in Fig2(c) are the (*MSE, L2-loss*) pairs obtained by in the trajectory of residual
gradient. Fig2(d) is similar, but plot the (*Bellman-error,  K-loss*) and (*Bellman-error,  L2-loss*) pairs. As mentioned in
L264–268, these scatterplots show good correlation between our K-loss and MSE/BE, suggesting it is a good proxy for
learning the value function.

[Meta-Review · NeurIPS 2019]

There is general consensus that the idea introduced in the paper is novel and interesting. Yet, I encourage the authors to read carefully the reviewers' comments and take them into consideration in the camera ready. In particular, the connection with the nested formulation of LSTD should be discussed to frame the contribution of the paper better.